# Co-Production at Work: The Process of Breaking Up Sitting Time to Improve Cardiovascular Health. A Pilot Study

**DOI:** 10.3390/ijerph19010361

**Published:** 2021-12-30

**Authors:** Thomas D. Griffiths, Diane Crone, Mike Stembridge, Rachel N. Lord

**Affiliations:** Centre for Health, Activity and Wellbeing Research (CAWR), School of Sport and Health Sciences, Cardiff Metropolitan University, Cardiff CF5 2YB, UK; DMCrone@cardiffmet.ac.uk (D.C.); mstembridge@cardiffmet.ac.uk (M.S.); rnlord@cardiffmet.ac.uk (R.N.L.)

**Keywords:** co-production, physical activity, sitting time, occupational health, cardiovascular health

## Abstract

Prolonged sitting negatively affects several cardiovascular disease biomarkers. Current workplace physical activity interventions to reduce sitting result in inconsistent uptake and adherence rates. Co-production attempts to improve the translation of evidence to practice through engaging the participants within the intervention design, improving the context sensitivity and acceptability of the intervention. A needs analysis questionnaire was initially conducted (*n* = 157) to scope workplace behaviours and attitudes. A development group (*n* = 11) was consulted in focus groups around the needs analysis findings and asked to comment on the feasibility of a proposed intervention. A pilot intervention was then carried out (*n* = 5). The needs analysis indicated that only 1.8% (*n* = 4) engaged in occupational physical activity, and 68.7% (*n* = 103) sat for ≥6 h during their working day. Through the focus groups, an intervention breaking up sitting time hourly with five-minute walking breaks was co-produced. Cultural and pragmatic issues concerning the implementation of frequent physical activity breaks from sitting and the subsequent impact on work productivity were highlighted. The pilot intervention increased the number of breaks from sedentary behaviour from 2 to 11. The co-production methodology resulted in a research- and stakeholder-guided compromise. Large-scale intervention implementation is required before firm effectiveness conclusions can be made.

## 1. Introduction

The accumulation of sedentary behaviour is a significant public health concern, reflected by the recommendations from the World Health Organisation [1,2]. By definition, sedentary behaviour relates to actions which incur less than 1.5 metabolic equivalents such as sitting [3], an ever-more prevalent behaviour in the workplace [4]. On average, adults spend over eight hours of their waking day in a sitting position [5,6,7], and half of this time is performed in prolonged unbroken bouts of sitting for over 30 min [8].

An ever-growing body of research has indicated that prolonged sitting time, defined as uninterrupted sitting for one hour or longer [9], is associated with an increased risk of developing cardiovascular disease [10,11,12,13,14,15,16], irrespective of meeting physical activity guidelines [17,18]. More specifically, it has been reported that prolonged sitting adversely affects cardiovascular disease biomarkers such as blood pressure [15,19], waist circumference [10], total cholesterol [12,13] and postprandial glucose [11,14], despite meeting physical activity guidelines [17,18]. Each additional hour of sitting increases the odds of developing type II diabetes and metabolic syndrome by 22% and 39%, respectively [20].

Office-based workplaces, where prolonged sitting is common, are considered a prime setting for public health interventions [21,22]. The first attempts to reduce occupational sitting were ineffective, citing methodological and study implementation issues [23]. More recent attempts have incorporated the use of active workstations and participatory approaches to better incorporate the organisational, environmental, and personal considerations to reduce sitting time [24,25,26,27]. Specifically, Healy et al. (2016) reported that the use of a sit–stand desk led to reductions in sitting time at three and twelve months [28]. This observation was supported by similar cluster randomised controlled trials [25,29] and meta-analyses [30]. Moreover, as well as sitting time reductions, Edwardson et al. (2018) reported improvements in job performance, work engagement, occupational fatigue, sickness presenteeism, daily anxiety, and quality of life through the implementation of sit–stand desks, alongside organisational, environmental, and motivational supports [25]. However, active workstations are not without their acceptability and feasibility issues, such as cost and specific effects on work-based tasks which limit their implementation [24,31]. Moreover, whether the reduction in sitting as a result of increased standing time leads to an attenuated or improved cardiovascular biomarker profile is less clear [32,33,34]. For example, Mantazri et al. (2019) observed no changes in blood pressure, waist and hip circumference, weight, body fat percentage and cholesterol, whereas Bodker et al. (2021) observed significant changes in triglycerides [32,33]. Similarly, Graves et al. (2015) reported that eight weeks of sit–stand implementation led to a decline in total cholesterol, but not plasma glucose or plasma triglycerides, nor systolic or diastolic blood pressure [34]. Consequently, interventions which result in greater increases in physical activity and yet remain feasible are required [35].

Breaking up sitting time with short, frequent bouts of physical activity has been shown to improve the cardiovascular disease risk profile [36,37,38] and cognitive function [39] in controlled laboratory trials. Studies implementing these physical activity break strategies in “free living” conditions over eight weeks have also reported good acceptability, feasibility and positive health effects [40,41]. Mailey et al. (2016) compared frequent short breaks to less frequent longer breaks. Interestingly, only the frequent short breaks condition reduced sedentary time and improved the cardiovascular disease risk profile [40]. Carter et al. (2020) used computer-based prompts to interrupt sitting time every 45 min with two-minute walking breaks, reporting an increase in sitting and standing bouts as well as strong retention rates [41]. However, it is important to note that financial rewards were offered by Carter et al. (2020), and Mailey et al. (2016) identified a number of barriers to participation and coping mechanisms [40,41]. This highlights the need for the inclusion of participants during the design phase of the intervention to maximise adherence and sustainability [42].

We utilised a co-production methodology to design a workplace intervention focused on breaking up sitting time with short, frequent bouts of physical activity to optimise uptake and adherence. Coined in the late 1970s for the purpose of the delivery of public services, co-production is a practice of delivery that engages multiple stakeholders in the process of intervention development, delivery, and its subsequent evaluation. By using a process of consultation and meaningful engagement with the intended participants, co-ownership of the intervention is cultivated by the participants, which results in improved context sensitivity and acceptability [43,44].

To date, co-production models have been adopted, with promising findings in physical activity related intervention development [45] and have shown favourable outcomes once implemented [46]. However, until recently, co-production has not been applied in a physical activity promotion context in the workplace setting. Mackenzie et al. (2021) utilised a co-production methodology to develop “Sit Less at Work” interventions across a range of organisations [47]. They detailed in-depth process and implementation data prior to subsequently evaluating the intervention’s effectiveness, reporting inconsistent findings [48]. More specifically, they reported a small decrease in workplace sitting time in small businesses but an increase for charity and local authority settings.

The aims of the current study were therefore to: (1) present the methodology and findings from the participatory co-development phase of workplace intervention focused on breaking up sitting time in a city, South Wales, UK; (2) provide an insight into factors that need to be considered when translating evidence to practice in a workplace setting; and (3) identify the challenges and facilitators of using co-production in participatory research involving multiple stakeholders.

## 2. Methods

### 2.1. Study Design and Context

Study procedures were approved by the Cardiff Metropolitan University School of Sport and Health Sciences Research Ethics Committee (approval code PGR-2903). The study took place in Cardiff, UK, between June 2020 and March 2021. At the time of the study, Wales was subjected to a national lockdown due to COVID-19, limiting travel, with the majority of people working from home or hybrid working. As such, the study was conducted mostly online, with limited in-person contact. The study used a mixed-method approach, including the use of an online survey to obtain a snapshot of current workplace practices and perspectives, focus group meetings to support the development of the intervention, and a feasibility trial. This approach was selected to explore the effectiveness of utilising the co-production methodology to develop a physical activity intervention focused on breaking up sitting time in office workers. This study also explored the feasibility of co-produced interventions in desk-based workers.

### 2.2. Participants

A voluntary sampling approach was used to obtain responses to an online ‘needs analysis’ questionnaire, posted on the social media platform Twitter and through local business contacts, from a variety of adult office-based workers (*n* = 164). The inclusion criteria were (i) adults (18 years +), (ii) occupied in seated job roles (defined as sitting for six hours or longer during working hours), and (iii) were physically inactive (defined as not meeting UK physical activity guidelines [1]). Exclusion criteria were (i) individuals who currently used active workstations (defined as reporting the regular use of sit–stand, treadmill or pedal desks), (ii) unable to complete any desk based focused physical activity, (iii) failed to occupy sedentary jobs, and/or (iv) met/exceeded UK physical activity recommendations [1]. Based on responses, seven participants were excluded because they failed to meet the inclusion criteria of occupying sedentary occupations and/or engaging over physical activity guideline recommendations [1]. 

A total of *n* = 157 office workers occupying desk-based occupations, including education/research, administration, human resources, accountancy, sales, and IT, were included for analysis. Based on responses to an email invite, *n* = 21 opted in for a follow-up focus group. A co-production development group (*n* = 11) was then recruited based on invite acceptance. It was important that the development groups contained a range of individuals to capture and aid organisational, environmental, and intrapersonal considerations. This group constituted a range of levels of employees (*n* = 3 management, *n* = 2 sales, *n* = 3 IT, *n* = 3 human resources) from six different organisations in four fields of employment (*n* = 2 public health, *n* = 5 IT, *n* = 3 energy supplier, *n* = 2 education). The intervention was subsequently piloted in a sub-section of participants (*n* = 5) from different levels of employment status (*n* = 2 management, *n* = 1 technician, *n* = 2 sales) at the same IT organisation. The role of the researcher in this study was to facilitate discussion, and where required, provide knowledge and evidence on the topic when participants asked for information or verification during the discussions. This discussion centred around workplace-specific knowledge and personal experiences to inform the development of the intervention.

### 2.3. Data Collection Methods and Process of Analysis

In line with recommendations from the Medical Research Council [49,50], a phased approach to the development of complex interventions was implemented between June 2020 and March 2021. This participatory process started with a needs analysis (online survey), followed by a co-production development phase. The development phase included three online participatory focus group meetings, including regular debriefing sessions with the research team between each meeting, before the co-produced intervention was subsequently piloted (Figure 1). The piloting phase ensured that the intervention was refined sufficiently before the delivery of a future large-scale effectiveness trial. Appendix A depicts the objectives and key tasks of each participatory stage.

#### 2.3.1. Development Stage 1: Needs Analysis—Online Survey 

The online survey was developed using the online platform Qualtrics and distributed through social media platforms and local business contacts (Cardiff, UK). Following informed consent, the online survey gathered participants’ demographics, current physical activity, and sedentary behaviour habits. More specifically, participants were surveyed about how many hours they engaged in occupation, leisure, transportation, and housework-based physical activity, as well as whether they had workplace provision (gym/equipment, e.g., cardio equipment, resistance bands, exercise space) for physical activity at their organisation and/or home-office. They were also surveyed about how many hours on average they sat during a working day. Through the online survey, participants were also asked about their current workplace physical activity opportunities and provision, before being questioned about their initial thoughts around different strategies to break up sitting time with physical activity. At the end of the online survey, participants were given the opportunity to “opt-in” to be a part of the development group. 

Data obtained from the online survey were grouped and coded for analysis. To avoid researcher bias, two other researchers also reviewed the data. Quantitative data relating to physical activity, sedentary behaviour, and preferences between the two proposed interventions were averaged to establish group consensuses. A thematic analysis was conducted with the qualitative data relating to the participants’ perspectives and opinions on workplace physical activity, sedentary behaviour, and the possibility of breaking up occupational sitting time with regular physical activity breaks. Data were discussed as a research team to resolve any disputes, prior to being summarised and used as focal discussion points within the focus groups.

#### 2.3.2. Development Stage 2: Eligibility and Intervention Framework—Online Participatory Meetings

Three focus group meetings were organised between June 2020 and March 2021 to facilitate the iterative development of the intervention. The focus groups were online due to the COVID-19 pandemic restricting face-to-face engagement. Objectives were pre-determined for each focus group meeting, as highlighted in Appendix A, but discussions allowed for an iterative approach between topics. The focus groups were run in small groups (*n* = 4–6) ensuring that the participants involved reflected multi-level occupations. Each meeting was facilitated by the researcher, who ensured that all participants were given a voice using open questions and asking for alternative viewpoints, specially from those that had not previously voiced an opinion. 

The first focus group took 95 min, with an online presentation and a designated note area which was publicly visible. During the first focus group, participants were educated on the importance of breaking up sitting time regularly for cardiovascular health before being asked to comment and discuss the concept of breaking up sitting time and the initial perceived challenges. Specifically, discussions centred around the current workplace culture being unaccepting of the notion of taking time out of work tasks to break up sitting time with physical activity. They were also asked to comment on and discuss how they would prefer to break up sitting time. The second focus group was conducted with the pilot group and lasted 65 min, with the use of an online flip chart for note taking. During this focus group, participants built upon the previous meeting by discussing and commenting on the details of the intervention, e.g., the intervention components (how often, how long for, which modality) and intervention support (e.g., daily email reminders). The intervention and the support were summarised in the conclusion to the focus group.

To facilitate as natural a discussion as possible, strategies were used to ensure participant interactions to adhere with the key principles of focus groups [51,52]. These strategies included asking participants to keep their cameras on, to aid identifying any non-verbal cues and responding through direct follow-up questions with their perspective. Additionally, participants were encouraged to use the hand-up or chat function while another participatory member was speaking. The researcher facilitated each meeting and data were collected through both audio recording and note taking. Data from audio recording (verbatim transcriptions), the visual prompts used (e.g., the PowerPoint slides used to guide the discussions) and researcher reflections were included in the summary of key points from each meeting. These were then shared with the participants to ensure that stakeholder views had accurately been interpreted, with no recorded disputes. These data were analysed deductively, and where topics arose that were unexpected or not linked to the deductive focus, these were noted. 

The development of the findings was complemented by frequent time-sensitive debriefing sessions with the research team to discuss developing findings, which were then used to inform the subsequent focus groups. The research team offered an outside perspective and suggested alternative avenues of discussion, firming attainable targets for the following focus groups. As such, an iterative and reflective practice process was adopted [36] and underpinned the development of the intervention to be piloted. 

#### 2.3.3. Development Stage 3: Intervention Piloting 

Through the co-production process, an intervention that broke up sitting hourly with five minutes of light-paced walking by the desk during working hours was designed. This was supplemented with support in the form of daily email remainders (prompts) highlighting the importance of breaking up sitting time. The pilot intervention started with one week of baseline physical activity and sedentary behaviour data collection followed by two weeks of breaking up sitting time hourly with five minutes of walking during working hours. During this three-week period, physical activity and sedentary behaviour were measured using the commercially available ActiGraph GT3x accelerometer (ActiGraph, Pensacola, FL, USA). Participants were instructed to wear the monitor on their non-dominant wrist during waking hours across the three weeks. Participants were also asked to note when their working day started and ended to help with analysis. The monitor was set to record raw triaxial acceleration at 30 Hz. Following completion of the intervention period, data gathered between working hours were downloaded to a computer using the manufacturer’s software (ActiLife software version 6.13.3; ActiGraph, Pensacola, FL, USA). Moderate and vigorous physical activity were defined using the acceleration levels >69.1 g and >258.7 g, respectively [53]. Breaks from sedentary time were defined as any instance where one minute identified as sedentary (e.g., counts per minute < 100) was followed by one minute identified as not sedentary (e.g., counts per minute ≥ 100) [54]. These data were later grouped, averaged, and analysed through a one-way ANOVA to compare intervention effects. Participants were also asked to note any challenges or facilitators of their engagement in the intervention.

#### 2.3.4. Development Stage 4: ‘Follow-Up’ Intervention Development—Online Participatory Meetings

To conclude the participatory research process and intervention development, participants who engaged in the intervention pilot met online for a follow-up focus group to discuss the challenges and facilitators of the intervention and intervention support. The meeting lasted 45 min, with the use of an online flip chart to openly note down the challenges and facilitators of the pilot intervention. Proposed solutions to challenges were openly suggested before concluding the participatory process and proposed intervention refinements. Based on the feedback, the intervention was refined in an attempt to improve the intervention acceptability and adherence. These refinements centred around the support mechanisms in place, including reducing the frequency of email reminders to once daily. The reminders contained key messages of the importance of breaking up sitting time for cardiovascular health. These changes were then communicated back to the pilot participants.

## 3. Results

The research had a clear objective to co-produce a workplace intervention to break up sitting time with physical activity; findings from the online survey and the participatory research process are presented in response to the overall objectives of the research.

### 3.1. Participant Characteristics

#### 3.1.1. Development Stage 1: Needs Analysis—Online Survey 

A total of *n* = 157 office workers occupying desk-based occupations, including research, education, administration, human resources, accountancy, sales, and IT, were involved in the study. Of the 157 office workers, 138 were female and 125 were working from home. The sample population represented individuals from a range of ages (18–24 *n* = 8; 25–30 *n*= 22; 31–40 *n* = 47; 41–50 *n* = 51; 51–60 *n* = 26; 60+ *n* = 3). 

#### 3.1.2. Development Stage 2: Eligibility and Intervention Framework—Online Participatory Meetings

The focus group sample (*n* = 11) was ensured to be representative of the needs analysis population. Thus, the focus group sessions were designed so that, where possible, there was equal male and female representation (*n* = 7 females) as well as equal representation of different job roles. 

#### 3.1.3. Development Stages 3 and 4: Intervention Piloting and Follow-Up Focus Group

Of the focus group sample, *n* = 5 participants (average ±SD age: 30 ± 5, three female) all from the same organisation, completed the three-week pilot intervention and attended the follow-up focus group. The employees held management (*n* = 2), HR (*n* = 1), sales (*n* = 1), and IT (*n* = 1) positions.

#### 3.1.4. What Factors Must Be Considered when Translating Evidence to Practice in a Workplace Setting?

Current Workplace Perspectives:

It was apparent from the needs analysis that participants’ physical activity generally took place outside of working hours (Figure 2A), with participants citing a “lack of time” during work hours for physical activity. There was an acceptance and understanding of the rationale for physical activity at the workplace, although there was a marked absence of workplace physical activity provision (Figure 2B). Moreover, participants reported long durations of workplace sitting (Figure 2C). During discussions on breaking up sitting time, participants highlighted the importance of breaking up prolonged sitting time and considered it novel. Participants were not familiar with an understanding of the negative impacts of prolonged sitting, nor that frequent breaks from sitting were advocated alongside physical activity guidelines. It was evident from the needs analysis that for the behaviour change intervention to be successful, it needed to be underpinned by education and a change in social norms.

#### 3.1.5. Effectiveness of the Workplace Physical Activity Intervention Pilot

Overall, the number of breaks from being sedentary significantly increased from 2 prior to the intervention, to 10 (*p* = 0.01) and 11 (*p* = 0.04) in weeks one and two, respectively (Figure 3A). Moreover, compared with prior to the intervention, there were no changes in daily average calories expended (kcal; Figure 3B; *p* = 0.642) or average daily moderate–vigorous physical activity (Figure 3C; *p* = 0.583). Across the three weeks, average MVPA per day fluctuated from 100 min in week one to 109 and 89 min in weeks two and three, respectively.

#### 3.1.6. What Are the Challenges and Facilitators of Conducting Participatory Research Involving Multiple Stakeholders?

Through these discussions, the co-production process highlighted the lack of congruence between the perceived ‘ideal’ and the feasible. A summary of the perceived facilitators and challenges of the participatory research process are highlighted in Table 1.

## 4. Discussion

### 4.1. Main Findings

The purpose of this study was to co-produce a sitting break intervention and test the feasibility of breaking up sitting time hourly with walking. This study highlights the facilitators and challenges of co-producing a workplace physical activity intervention focused on breaking up sitting time. The process provides a successful example of co-production in designing a workplace physical activity intervention and highlights the need for cultural (movement towards it being acceptable to take regular computer breaks) and educational (awareness of the importance of doing this for health) shifts to overcome many of the barriers to participation. Specifically, there was a clear lack of knowledge about the importance of breaking up sitting time, highlighting the disparity between guidelines and dissemination. Moreover, it was repeatedly highlighted that not only the intervention needed to prevent distraction from work, but employer and management endorsement and engagement was important in relation to changing perspectives of social normality and employee engagement in the intervention. Lastly, during the focus group discussion, it was evident that an intervention focusing on half-hourly breaks, where a bulk of the breaking up sitting time laboratory studies are focused [14,35,55,56], would be unfeasible due to the frequency of the disruption to work tasks and meetings often lasting an hour. Therefore, an hourly break strategy was implemented as a result of the compromise between the stakeholder input and researchers’ evidence-focused approach.

The pilot intervention illustrated that the intervention was successful in breaking up sitting time frequently, as exemplified in the breaks significantly increasing from sedentary time: from two breaks pre-intervention to eleven breaks in week two (*p* = 0.04) of the intervention with no adherence issues reported. This is of specific significance considering the importance of breaking up sitting time in a frequent manner on both cardiovascular biomarker outcomes [14,16,17,36,37], as well as the potential cerebrovascular outcomes [57]. However, the physical activity breaks did not impact caloric expenditure or moderate-to-vigorous physical activity (MVPA; Figure 3b,c) during working hours. It is conceivable that MVPA was not altered in the intervention, considering that walking may only be light-intensity exercise. 

### 4.2. What Does This Study Add?

Following the work by Mackenzie et al. (2021), this is the second study to apply co-production to the development of a workplace physical activity intervention [47,48]. This study provides insights into the factors that should be considered when translating evidence to practice, and, in particular, the importance of tailoring the intervention to the participants’ needs and capabilities with competing organisation and cultural demands. Buckley et al. (2018) highlighted similar key facilitators and challenges of participatory research in a physical activity promotion context. For example, through the ‘needs analysis’ process and subsequent reflective practice in both the current study and Buckley et al. (2018), changes in cultural norms within the workplace were identified as a requirement for interventions to succeed [45]. 

There are a number of factors that influence the success of physical activity workplace interventions, including: participant characteristics, study design, and intervention rationale [41,58]. To date, some studies have employed participatory approaches alongside active workstations to better incorporate the organisational, environmental, and personal considerations to reduce sitting time, with apparent success [24,25,26,27]. However, active workstations are not without their acceptability and feasibility issues, which potentially limits their use [24,31]. Free-living physical activity modalities such as walking breaks have received encouraging success in breaking up sitting time and improving both cardiovascular biomarkers [41,59,60]. Therefore, through co-producing a free-living physical activity intervention, we aimed to overcome these potential barriers whilst incorporating the stakeholders within the design and delivery of the intervention, and senior management support to create appropriate social norms within the employment setting. Through the co-production process, it was evident that stakeholders shared a notion of the importance of physical activity and breaking up sitting, in line with the extant evidence, although they also indicated the importance of minimising the disruption to their work tasks, and more specifically, their workflow. Through the focus groups, it became evident that the preferred scientific approach, incorporating breaks every half an hour, would not be feasible for translating this into the workplace. As such, an evidence-based compromise towards hourly breaks was adopted [38,61].

The stakeholders involved in this process were not aware of the health implications of sedentary behaviour; specifically, the importance of regular sitting interruptions [61]. Although national guidelines on physical activity have been published, which also include recommendations for reducing sitting time [1], there is also an apparent disparity between evidence and public awareness, highlighting the importance of an initial educational component within physical activity interventions [41]. The notion of establishing a policy for taking breaks from seated behaviour was mentioned within the initial development group, stressing the requirement for infrastructure to support the activity at the workplace.

Co-production is founded on the principle that those who feel ownership and autonomy in their physical activity are more likely to have improved long-term adherence [62]. This method therefore affords the potential for the translation of public health messages into actionable and sustainable behaviour changes. In practice, co-production requires considerate and transparent negotiations to enable progressive actions which are aligned across all stakeholders [63]. We found that by starting with an initial ‘needs analysis’, we were able to identify challenges at an early stage before proposing, discussing, and modifying the final concept. The development groups allowed for facilitated discussions and refinement of the initial concept intervention, which resulted in an agreed evidence-based intervention which is deemed acceptable and feasible by the stakeholders.

### 4.3. Limitations

The purpose of this study was to detail the co-production approach to intervention development. However, there are limitations to this approach: (i) Inconsistent stakeholder attendance meant that not all stakeholders were available to provide input to the discussions of all meetings; (ii) The focus group meetings were conducted online rather than in person due to COVID-19 restrictions; as such, the level of interaction was somewhat limited; and (iii) Aligning the research to reality required compromise from both the stakeholders and research group, not always guaranteeing resolution for all stakeholders; (iv) A small but representative sample of participants took part in the development (*n* = 11) and piloting (*n* = 5) of the intervention, which although allowed for more in-depth discussion and greater participant input, could have impacts on future applicability, adherence, and heterogeneity. Moreover, although the intervention was designed with a larger group of participants from a number of different organisations, it is unknown whether the intervention translates to other organisations. Future implementation of this intervention across a 12-week period will better characterise the interventions’ effectiveness on improving cardiovascular and cerebrovascular biomarkers, as well as participant adherence to the intervention; (v) Another limitation within data collection is the wrist placement of the accelerometer to gather sedentary data. Previous studies have indicated a small overestimation of sedentary breaks from sensors on the wrist [64]. However, because this wrist placement was consistent across the pilot, this is unlikely to have affected the outcomes; (vi) Lastly, in an attempt to understand the environmental factors that could support regular breaks, participants were questioned during the needs analysis about their sitting break provision at both their workplace and home-office. Although many physical activity opportunities, such as stair use and printer trip breaks were identified, these strategies were not explicitly measured, potentially under-representing the provision of sitting breaks at the workplace site.

## 5. Conclusions

To the best of our knowledge this is the second study to describe the participatory, co-production process of a workplace physical activity intervention. By adopting a co-production model, we were able to include the participants’ views and opinions within the intervention design, improving the context sensitivity of the final designed intervention. Pilot work suggests that our intervention is successful at reducing sedentary time over a short duration; however, the long-term effectiveness of the intervention is yet to be determined. There are a number of important facilitators and challenges when implementing a physical activity intervention into a workplace setting that should be considered through the co-production process.

## Figures and Tables

**Figure 1 ijerph-19-00361-f001:**
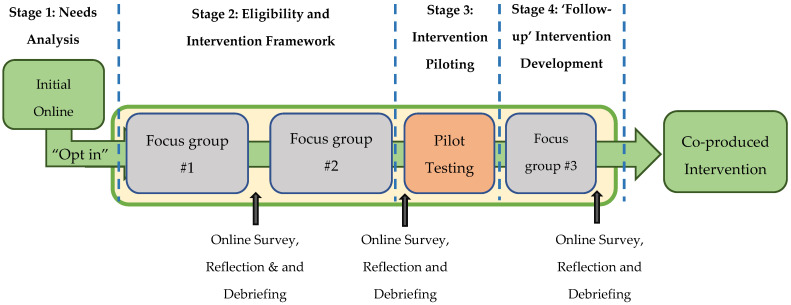
An overview of the participants’ journey through the participatory process.

**Figure 2 ijerph-19-00361-f002:**
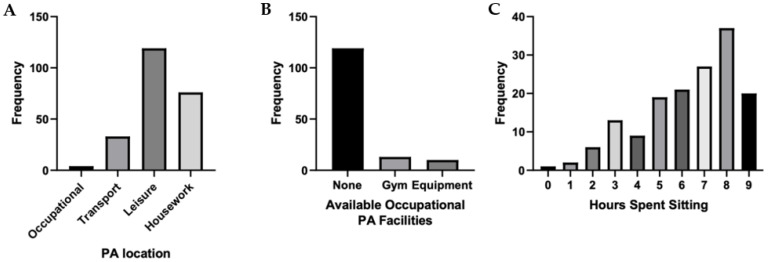
A view of (**A**) physical activity behaviours of office-based workers, (**B**) the current organisation workplace physical activity provision, and (**C**) the average time spent sitting down at work based on self-reported data (*n* = 157).

**Figure 3 ijerph-19-00361-f003:**
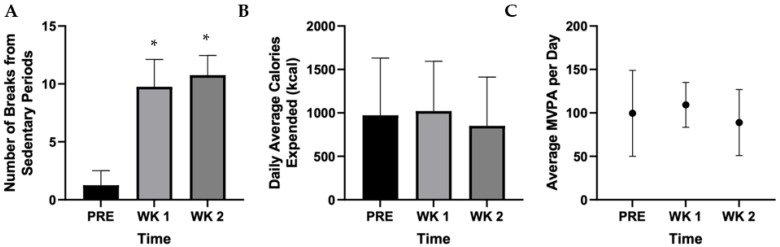
Pilot accelerometer data for (**A**) the change in number of breaks from sedentary time, (**B**) the change in daily average calories expended (kcal), and (**C**) the change in MVPA per day *n* = 5. * Indicates statistical difference compared with the baseline (*p* < 0.05).

**Table 1 ijerph-19-00361-t001:** Summary of the facilitators and challenges of a participatory research process.

*Facilitators*	*Challenges*
Using the initial questionnaire and subsequent follow-up focus group as a ‘needs analysis’ allowed for the stakeholders to share their perceptions both individually and as a group; By using the initial questionnaire within the needs analysis, we were able to acquire a large number of stakeholders at different levels and their perspectives with an ability to gain those in both public and private sectors; Having a structure in the focus groups which allowed for open discussion guided by open questions, facilitated input and discussion from stakeholders to share their views, knowledge, and experiences to help inform the intervention; Following focus groups, consultation back with the research group allowed for an outside expert perspective, which would suggest following discussion directives; Reflective practice facilitated intervention development and knowledge translation into meaningful action points; The pilot intervention allowed for the development group to trial their intervention to test the structure and feasibility of it in their working days; The follow-up focus group after the pilot allowed for actionable refinements to the intervention to improve the intervention.	Although the development meetings were attempted to work around stakeholder availability, irregular stakeholder attendance meant a loss of input; Aligning the research with reality required compromise from both the stakeholders and research group, not always guaranteeing resolution; Aligning the perception with the reality of the physical activity intervention. Participants often expressed concern that the intervention was of a higher intensity than it actually was; Aligning the focus of breaking up sitting time frequently alongside the culture of workplace. There was reluctance to participate with conflicting tasks such as online/in-person meetings; Challenges highlighted the importance of support from colleagues and management and the need to not feel judged by peers; The perception of how breaking up sitting time would impact on interrupting workflow and productivity; The impact of the quantity of workload on the participants perceptions of being able to carry out the intervention; The importance of physical activity and breaking up sitting time during the day, especially during working hours where the focus is largely on deadlines.

## Data Availability

The datasets used and/or analysed during the current study are available from the corresponding author on reasonable request.

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
