# Peer review of "Co-Production at Work: The Process of Breaking Up Sitting Time to Improve Cardiovascular Health. A Pilot Study"

_ijerph, 2021, doi:10.3390/ijerph19010361_

Round 1

Reviewer 1 Report

In the study entitled “Co-production at Work: The Process of Breaking up Sitting Time to Improve Cardiovascular Health”, the authors conducted a needs analysis questionnaire and performed a pilot intervention to reduce sedentary time during working hours. Major revisions are needed.

The introduction does not provide sufficient background and relevant references. Important definitions (e.g., sitting time, sedentary time) are needed.

Materials and Methods section must be improved. Study design part does not report any information about the design.

Type of study is unclear.

Methods and algorithm used to obtain physical activity (ActiGraph) and sedentary time must be reported. Also, the type of accelerometer must be declared.

The results are not clearly presented. Are the results reported in figures 2,3 statistically significant?

Description along the test and graphs are not correctly linked (e.g., line 181,182 do not refer to figure 3).

Do the authors a power analysis to check if the intervention sample (n = 5) was sufficient? If so, please insert the power analysis.

Discussion should be improved including relevant references and better discussing main findings of the study.

Reviewer 2 Report

This article described the co-production of a workplace intervention targeting increasing the number of breaks in sitting time. The major concern with this article is the lack of detail throughout, and the insufficient acknowledgement and reference to the literature in the field. Specific comments are below.

  1. Introduction: Lines 28-30: Recommend updating references to include the umbrella reviews relating to the inclusion of sedentary behaviour recommendations into physical activity guidelines by (for example) USA, Canada and the WHO.
  2. Please provide a reference for the statement on line 34 “a physically active workforce is associated with improved 34 productivity and less sick-related absence,

  1. Introduction, lines 35-37. The authors state that “evidence suggests that current workplace physical activity interventions are ineffective at altering physical activity or sedentary time.” It is unclear what the authors are including as “physical activity” interventions, as several interventions that have targeted both sitting less and moving more have demonstrated substantial reduction in workplace sedentary time – including through encouraging regular breaks in sedentary time. These interventions have also been evaluated via large, cluster randomised trials (see for example the Stand Up Victoria Study (Healy et al., MSSE 2016), The Stand and Move at Work study (Pereira et al., IJBNPA 2020) and the Stand More at Work study (Edwardson et al., BMJ 2018). These studies have also used participative approaches to identify the intervention strategies they will use within their teams. Other studies have also described iterative intervention development of sit less / move more interventions (e.g., Neuhaus et al., IJBNPA 2014) and the partnerships between government and researchers to adapt research into practice (e.g., Healy et al. AIMS Public Health 2016). It is recommended that the authors acknowledge these larger trials (and the associated systematic reviews that examine the various strategies e.g., Shrestha et al., Cochrane Reviews 2018) and prior evidence in the introduction. There is novelty in the description of the co-design process; however, it is important that the authors accurately review and summarise the broader field of research.

  1. Methods: It is recommended that the sampling and recruitment methods, as well as inclusion and exclusion criteria, are included in the participant section. Please also make clear the number of organisations that participants were from. Some of the information included in the methods seems more appropriate for the results (e.g., descriptives of participants). It is unclear on the recruitment to the sub-groups. Were all chosen? (i.e., did only 11 opt in?) How were the five for the pilot testing chosen? Were they from different organisations? Did they have different experiences? Please provide more detail on this process. It is also unclear if the workplace / organisation had any input into the development of the intervention. Behaviour change is influenced by multiple factors (e.g., organisational, policy, cultural, environment, interpersonal) and addressing change only at the individual level is unlikely to lead to sustained success. It would be good to see more detail on how the authors considered this multiple-level influences.

  1. Methods: needs analysis – please provide details, including measurement metrics, and references for the measures used. Also provide more detail on the qualitative analysis – how many coders were used? How were disputes settled?

  1. Methods: focus groups. As per previous comment, please provide more detail on how those for the focus groups were selected, and the characteristics of those that took part. Were the reflective of the broader sample from the needs analysis?

  1. Intervention piloting: more detail is needed here, including on the Actigraph monitors (what position were they worn, when were they worn [24 hours?], how were work hours measured? how were the data processed? How were participants chosen for this piloting? How was the sample size (n=5) determined? What were the actual questions asked? There is also no description at all either here or in the results of the actual intervention. What were participants supposed to do? How was the feedback that education and cultural norms were important fed into the intervention development?

  1. Stage 4 – how long was this meeting? Were participants informed of what changes were made?

  1. Results: line 171 – the detrimental health impacts of too much sitting are not irrespective of how much physical activity – high levels of physical activity can counteract detrimental impacts. Please modify this statement. It would also be useful to have quotes accompany the qualitative interpretation, and also identification of what were the main themes.

  1. Results: detail is needed on the characteristics of those that took part in the pilot, as well as all the methodological detail previously flagged.

  1. Figure 3: It is unclear how many people have contributed to this data, or where this data is coming from (self report? Actigraph?). Figures need to be able to stand alone.

  1. Without more detail on who took part, it is difficult to understand who the various stakeholders were.

  1. Discussion: as flagged in previous comments, several other interventions have used co-production elements in their designs. Please modify this statement.

  1. Discussion: the discussion includes elements of what the intervention involved that are not discussed elsewhere in the methods or results. As flagged previously, there should be a clear description of what the final intervention actually was.

Reviewer 3 Report

Summary

The purpose of the study was to use co-production to develop a physical activity intervention in the work place. Surveyed office-based workers then a sub-set helped with development (focus groups) then a sub-set of that group helped with piloting (testing program). Co-production was considered a success for developing an intervention.

Study Importance

Giving a group a say in an interventions development could assist with enjoyment, adherence, and practicability especially when that intervention is to occur in the work place.

General Comments

Very few participants took part in the development (11) and then piloting (5). Could this have a big impact on the possible applicability, enjoyment and future adherence to this program by others due to a lack of heterogeneity?

The figure titles for figure 2 and 3 do not match what is being shown in the respective images.

What is the intervention that was piloted/performed for 2 weeks? Were the breaks equally spaced throughout the day? How long were the breaks? What was done during the breaks? Did they go for a walk, do some stairs, just stand and work? This really needs more details. Why did the breaks go up from 10-11? Were more breaks suggested during week 2? What was the adherence to the pilot?

Doesn’t seem like the work breaks impacted caloric expenditure, so why would someone be inclined to break up sitting time based on the results from the piloting intervention?

I recognize the possible benefits from using co-production to develop an intervention for those who will be doing the intervention but the intervention developed for this pilot did not seem to have an impactful, measurable outcome (no change in caloric expenditure, no change in MVPA; in fact the means for week 2 go down…) other than successfully breaking up sitting time based on the current results provided.

Only address the viability of the co-production and consider it successful because sitting time was broken up in the discussion. Did not address why caloric expenditure and MVPA did not change.

Specific Comments

Figure 1: seems to be something in the co-produced box? I see some black markings below produced. Some previous text that was meant to be completely hidden?

Line 148-149: No statistics (parametric, non-parametric, effect sizes, etc.) performed on the data collected from the ActiGraph?

Line 219: So longer breaks were used in the pilot investigation? What constitutes a long break? Needs to be addressed in the methods.

Line 230: did the participants in the intervention portion have adherence within this pilot?

Round 2

Reviewer 1 Report

We checked the authors' corrections. Now, we confirm that the manuscript can be accepted.

However, English language must be revised.

Author Response

We checked the authors' corrections. Now, we confirm that the manuscript can be accepted However, English language must be revised.

We thank the reviewer for their time reviewing our manuscript. In relation to the reviewer’s comment “English language must be revised”, we have spent some time going over the manuscript to improve how the manuscript reads.

Reviewer 2 Report

Thank you to the authors for addressing many of the comments and concerns raised. There are still a number of concerns, as highlighted below. This includes how it is unclear how some of the issues raised in the co-production process (around social norms and management support) were actually addressed in the development of the intervention.

Abstract: Line 9: Recommend removing emotive and false statement “However, current workplace physical activity interventions to combat sitting report notoriously poor uptake and adherence rates.” Interventions are not about “combatting” sitting, it is about ensuring the right balance. I would also argue that there can be strong uptake and adherence rates of sitting less, moving more interventions.

Introduction: Line 32 A more updated reference for sedentary behaviour is required rather than [3] (there is now the sedentary behaviour research network consensus definition); this sentence should also include a reference for the statement in the same sentence regarding that it is “an ever more prevalent behaviour in the workplace”.

Introduction: Line 68: need a reference for the statement “Consequently, interventions which result in greater increases in physical activity or sedentary time and remain feasible have been called for.” Who has called for them?

Introduction: Line 87  - typo: “focused on” not focused of.

Introduction: Line 87 – the research cited highlights the good acceptability and uptake of previous studies, so it is unclear what issue regarding these factors are to be “overcome”. Please reword.

Introduction: Line 96. The authors state that co-production has not been used in the workplace, however a recent paper in IJERPH (published July) examined this question (https://pubmed.ncbi.nlm.nih.gov/34360045/). Please revise this statement.

Introduction: the authors refer to this as a physical activity intervention; however, the literature review and much of the argument is around prolonged sitting. It is recommended that the terminology reflects that (i.e., that is about regularly breaking up sitting time).

Methods: Line 200 – typo (lend to – unsure what is meant here).

Methods: Line 239 – please include the measurement metrics (i.e., reliability / validity) for the wrist worn monitor for detecting breaks in sedentary time.

Methods: Line 259 – what intervention changes were suggested?

Results: Line 295 – it is unclear what is meant by the statement “It was deemed that there was a requirement for behavioural change underpinned by education and a change in social norms.” Deemed by who? Do you mean that the behaviour change intervention should be underpinned by education and changes in social norms? How did you intervention then address these considerations in the design? In particular around the social norms?

Results: Are Figures 2 and 3 in the wrong order? For the pilot study, the MVPA seems pretty high (around 100 mins/day). This should be noted in your results.

Results: In terms of occupational PA facilities – did this relate to their workplace, or where they were currently working (considering most were working from home). What was meant by equipment in this context? What about other environmental factors that could support regular breaks? (e.g., centralisation of printers; policies for no lunch at desks; availability of both standing and sitting furniture; accessible, safe and attractive stairwells; interconnected walkways). If these were not measured, then this should be noted in your discussion.

Discussion: line 317 – I would argue that your aim was to co-produce and intervention to support workers to break up their sitting regularly; not to co-produce a physical activity intervention.

Discussion: Line 322 – the “need for cultural and educational shift to overcome many of the barriers to participation” was not explicitly clear in the results or in how the intervention was developed.

Discussion: Line 340 – as noted earlier, this is not the first study to use co-production.

Discussion: Line 396 – it is also important to note that the pilot study was conducted with participants from the one organisation. It is unclear if this same intervention could be used in different organisations. Are you recommending that a full co-production approach is used in every workplace? What about a participatory approach, where participants are guided from a needs analysis and collective decide on strategies, which has been shown to work across multiple organisations? (see the BeUpstanding program of research: www.beupstanding.com.au).  

Discussion: Line 399 – why a 3 week period?

Reviewer 3 Report

I would like to thank the authors for the time they spent revising their manuscript. It is clear you put in a lot of work to address the reviewer comments.

Given the small sample size for the intervention I recommend adding "A Pilot Study." to the end of the title. "Co-production at Work: The Process of Breaking up Sitting 2 Time to Improve Cardiovascular Health. A Pilot Study" I recognize the constraints imposed on research at the start of the pandemic but also given the lock downs in place around the world, many likely increased their sedentary time from working at home as well.

Author Response

I would like to thank the authors for the time they spent revising their manuscript. It is clear you put in a lot of work to address the reviewer comments.

Given the small sample size for the intervention I recommend adding "A Pilot Study." to the end of the title. "Co-production at Work: The Process of Breaking up Sitting 2 Time to Improve Cardiovascular Health. A Pilot Study" I recognize the constraints imposed on research at the start of the pandemic but also given the lock downs in place around the world, many likely increased their sedentary time from working at home as well.

We’d thank the reviewer for their time reviewing our manuscript. We acknowledge the comment that the reviewer has made and have edited the title to include “A Pilot Study”.